# E-Business Strategy in Developing Countries: A Framework and Checklist for the Small Business Sector

Martin Wynn [1,*] and Olakunle Olayinka [2]

1   School of Computing and Engineering, University of Gloucestershire, Cheltenham GL50 2RH, UK
2   Department of Computer Science, University of Sheffield, Sheffield S10 2TN, UK; o.olayinka@sheffield.ac.uk
*   Correspondence: MWynn@glos.ac.uk

**Abstract:** Since the turn of the century, there has been a growth in the use of e-business by both large and small companies worldwide, a trend that has been given further impetus by the move to online trading in the COVID-19 pandemic era. For small companies, there are the potential benefits of increased efficiencies and market share gain, associated with the re-engineering of selling and marketing processes; but in developing countries, such as Nigeria, research into how small businesses are using e-business systems and technologies is limited. This article builds upon earlier case study research in the Nigerian small business sector to develop a framework for e-business strategy development, implementation and review. Using an inductive approach, data was collected from six small businesses, using interviews and questionnaires, to profile the e-business operations of these companies. This study found that e-business strategy was generally lacking in these companies, but interview material was used to support the development and validation of the strategy framework, which provides a process and a checklist for small businesses pursuing e-business initiatives in developing world environments.

**Keywords:** e-business; Nigeria; small business enterprises; strategy development; strategy implementation; strategy framework

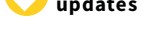



## 1. Introduction

The need for a properly documented plan and strategy to aid the successful deployment of e-business technologies is generally recognised. Chen, Ruikar and Carrillo [1], for example, suggest that organisations need to take a holistic approach to e-business implementation, which encompasses careful consideration of the expected benefits, which technologies to adopt and management roles. In a similar vein, Chaffey [2] concluded that, without a clearly defined e-business strategy, the adoption of e-business in organisations will often result in a waste of resources, poor integration of e-business and back-end systems, missed opportunities and ultimately, suboptimal business performance.

Nigeria is the largest economy in Africa, and the use of the internet, social media and mobile phones has grown rapidly in recent years, with internet penetration increasing at about 10% every year [3]. This has been the catalyst for increased awareness of, and demand for, e-business services and products. The adoption of e-business initiatives by Nigerian companies, however, is generally still in the early stages of development. In this context, this article puts forward a framework to support small business enterprises (SBEs) in their e-business initiatives. SBEs can be defined as enterprises that employ fewer than 50 people. The article also makes reference to research conducted in small-to-medium sized enterprises (SMEs), which are defined as having less than 250 staff [4].

In the broader context of digital transformation and sustainability, the implementation of new technologies such as e-business systems is seen by some as a route to improved efficiencies and sustainable growth [5]. Nevertheless, Antonio Guterres, United Nations Secretary General, has argued, "digital advances have created enormous wealth in record

time, but that wealth has been concentrated around a small number of individuals, companies and countries", and has warned, "under current policies and regulations, this trajectory is likely to continue, to further contributing to rising inequality. We must work to close the digital divide, where more than half the world has limited or no access to the Internet. Inclusivity is essential to building a digital economy that delivers for all" [6] (p. iv). Heeks [7], on the other hand, has highlighted how the spread of mobile computing, social media, cloud computing, and broadband in the developing world may provide sustainable solutions to seemingly elusive environmental and social challenges. Heeks [8] (p. 1) concludes, "we can foresee a 'digital development' paradigm in which ICTs are no longer just tools to enable particular aspects of development, but the platform that mediates development". E-business systems and associated digital technologies are already playing a significant role in this transition, ranging in scope, size and impact from the small-scale project initiatives considered here to the global e-commerce platforms such as Amazon, Jumia and Flipkart, which are major operators in several developing world economies [9].

The e-business strategy framework put forward in this article follows research into Nigerian SBEs undertaken in the 2015–2020 period [10]. In addition to a survey of over 50 small companies, six businesses were investigated in depth and their e-business activities were analysed and profiled. In addition, research results included the identification of critical influencing factors for e-business adoption. These results formed the basis for the development of the three-phase strategy framework presented here. The research objectives (ROs) for this study were thus two-fold: first (RO1), to combine an analysis of existing literature and the six case study profiles to build a framework to guide SBEs in e-business strategy initiatives; second (RO2), to validate the framework through semi-structured interviews and provide operational guidance as to how the framework can be deployed.

Following this introduction, the relevant literature, definitions and models are reviewed, and an outline of the conceptual framework and research questions are set out. In Section 3, the research methodology and design is discussed and the six case study profiles are presented. Sections 4 and 5 then address the two ROs. Section 4 presents and illustrates the framework, and Section 5 sets out feedback on the applicability of the framework from the company owners and other practitioners, providing some guidance for its use. Finally, Section 6 draws some conclusions regarding the contribution of the research project and future work.

## 2. Relevant Literature

### 2.1. The E-Business Concept

The term e-business was used by IBM in 1996 as part of their marketing campaign where it was defined as the transformation of key business processes through the use of the internet [11]. According to Chaffey [2], e-business generally aims to improve the effectiveness and competitiveness of an organisation by deploying IT equipment throughout the organization. Turban et al. [12] (p. 6) see e-business as "a broader definition of e-commerce, which is not just the buying and selling of goods and services, but also servicing customers, collaborating with business partner and conducting electronic transactions with an organization". Laudon and Laudon [13] (Section 1.1, para. 31) define e-business as "the use of Internet and digital technology to execute all of the activities in the enterprise. E-business includes activities for the internal management of the firm and for coordination with suppliers and other business partners", and this is the definition assumed here. Although some of the extant literature refers to e-commerce and e-business interchangeably, the definition of e-business adopted here assumes that e-commerce and online interaction with clients/customers are part of e-business, which is a somewhat broader concept.

### 2.2. E-Business Strategy in the Digital Era

The e-business strategy of an organisation defines the approach through which internal and external IT and e-business applications can support and influence corporate strategy. For most large organisations in developed countries, e-business initiatives are of-

ten driven by established e-business strategies and management support, which guide the implementation choices made, the e-business systems deployed, and provide a roadmap for e-business adoption, while ensuring alignment with the corporate strategy. In small businesses in the developing world, the existence of e-business strategies to guide e-business initiatives is less in evidence. The effort required to develop and implement strategy frameworks may be unacceptable to small businesses already constrained by limited financial and human resources [14].

The adoption of e-business can be seen as one element of what is often now termed "digital transformation", which refers to how the deployment of digital technologies can lead to new, disruptive business and value creation models [15]. However, simply digitizing the selling, marketing or procurement operations through technology alone will not necessarily benefit an organization, but rather a balanced interplay of new technology, people up-skilling and process re-engineering is needed, underpinned by a clear e-business strategy, which may be part of a broader digital transformation strategy [16].

Digital transformation encompasses a wide range of technologies and concepts that have been in evidence in the past five years or more. We may be at the dawn of a new technology age, which marks a step-change in our use of technology and a major change in business models, with the ramifications of the COVID-19 pandemic speeding up this change process. However, the development and implementation of e-business strategy will still be based on similar principles to those used in most strategy development fields, and many of the technology concepts and practices will also be based on well-proven antecedents.

*2.3. E-Business in Developing Countries*

Research conducted on e-business in developing countries has largely focused on factors responsible for adoption [17], barriers to adoption [18], challenges of adoption [19] and the benefits of e-business and consumer attitude to e-business adoption [20]. Relatively little has been published on e-business strategy per se, although the difference in cultural, technological and sociopolitical contexts between the developing and developed worlds is highlighted in some of these papers.

Rahayu and Day [17] attempted to establish the factors influencing e-commerce adoption in Indonesian SMEs. They studied 292 Indonesian SMEs using the Technological, Organisational and Environmental (TOE) model as an analytical framework. They concluded that there were 11 key factors influencing e-commerce adoption, which they classified as either technological, organizational, environmental or individual factors. Research into e-business in Nigeria has examined some aspects of e-commerce, but has failed to examine the transformational potential of e-business in re-shaping business processes and even underlying business models [21]. Nevertheless, Olatokun and Bankole [22] researched e-business adoption in SMEs in Ibadan, in southwestern Nigeria. Sixty SMEs were studied via questionnaires, and their findings suggested company size was of little relevance, but that the age of the company was, with younger companies being more likely to transition to e-business systems and processes.

Agwu [23] undertook case study research in six companies in three different regions of Nigeria, seeking to establish the factors influencing e-business adoption. The study found that consumer readiness, IT Skills shortage and internet connectivity are key factors in determining e-business adoption and effective website maintenance. Barriers to e-commerce adoption in Nigeria were similarly investigated by Agwu and Murray [24], who focused on the three states of Lagos, Abuja and Enugu. Based on qualitative interviews with the owners of SMEs, a number of inhibiting factors emerged: the absence of an e-commerce regulatory security framework, lack of in-company technical skills, and poor public infrastructure, including internet access.

Raghavan, Wani and Abraham [25] explored e-business adoption and trends in Indian SMEs. Similar to existing studies in other developing countries, their research identified owner-manager characteristics, technology factors, organisational factors and institutional

influences, as key issues affecting e-business adoption. Their study also recognised the impact that external pressure from industry, external suppliers and government could have on adoption. Abdullah, White and Thomas [26] also investigated the adoption of e-business in SMEs in Yemen. Their study focused on the e-business adoption pattern of SMEs by using a mixed method approach where data was gathered through more than 200 survey responses and five in-depth interviews. Findings from this study confirmed that SMEs progressed from email to more advanced e-business systems over time, but a significant number of SMEs began implementing e-business by deploying cloud services first.

### 2.4. E-Business Models and Frameworks

The internet has dramatically changed the way information is communicated, and this has transformed the conduct of business in several industry sectors [27,28]. New business models now exist which offer both consumers and organisations additional benefit. In this context, the e-business adoption process is portrayed by some to be a sequential process, as evidenced in a number of maturity models—for example, the Connect, Publish, Interact, Transform (CPIT) model [27] and the stage model for e-business development [29]. Other researchers have argued that e-business adoption is more complex, and reducing it to sequential stage models does not adequately reflect reality [30].

Today, e-business systems are becoming prevalent in SBEs due to the reduced cost of IT infrastructure and general advancements in technology. Deployment modes and approaches vary from one organisation to another. For some, it is simply the deployment of brochure websites for marketing purposes, while for others, it is the application of systems to effectively engage customers [31]. In addition to helping companies increase market share and grow their customer base, it is becoming increasingly evident that a clearly defined and effectively executed e-business strategy will deliver efficiency gains and other benefits for an organization [32].

The DTI Adoption Ladder was one of the early e-business frameworks designed in the United Kingdom by the Department of Trade and Industry, to measure the level of e-business adoption in SMEs [33]. This framework breaks down e-business elements into five sequential steps, which are: email, website, e-commerce, e-business and transformed organisation. This framework suggests that SMEs progress over a series of stages in a well-planned, progressive and sequential process from the use of email through to the development of a website, selling and payment online, integration of internal processes and external partners and finally, the transformation of the entire business. The CPIT model developed these concepts further to facilitate an assessment of e-business adoption at individual process level, rather than across the organisation as a whole [33]. This model offers a 2-dimensional matrix to evaluate the impact of e-business technologies across all main business processes over four stages: Connect, Publish Interact and Transform. Chaffey [29] put forward an e-business framework for organisations to assess their current use of e-business. Again, there are four stages in the model (web presence, e-commerce, integrated e-commerce and e-business), and four aspects of e-business (service availability, organisation scope, transformation type and strategy deployed) against which organisations can evaluate their position across the four stages.

Abdullah, White and Thomas [26] proposed an e-business "measurement evolution" model, which contains nine sequential stages through which small businesses deploying e-business are expected to pass. The model is based on the DTI adoption ladder; however, it introduced social media, mobile applications and cloud services, to cater for new technologies that have become widely adopted by SMEs since the development of the DTI adoption ladder. By using this model, businesses are expected to progress sequentially, starting at email, with the overall aim of becoming a transformed organisation. Further research by Abdullah, White and Thomas [34] applied this model in Yemen, indicating that businesses could have two points of entry: stage 1 (email) or stage 6 (cloud services). Businesses that began adoption from stage 6 (cloud services) could either progress to stage 8 (transformed organisation) or retrospectively deploy other technologies.

Chen, Ruikar and Carrillo [1] developed a holistic e-business framework for e-business strategy formulation and implementation in the construction industry. The authors advocate the need for multiple elements to be considered in order to adequately develop an e-business strategy and a roadmap for the construction industry. By combining various aspects and approaches from already existing models and framework, the model was designed for senior management staff to define e-business strategies and implementation plans. The model suggests six sequential stages—access situation, establish vision, define critical success factor, develop action plan, implementation and review. It also takes into consideration people, technology, process and management capabilities that are required for the successful development of an e-business strategy in these stages.

In a broader context, Heeks' studies on information systems in developing countries [35] similarly identified four aspects of an organization that must change as it adopts new technologies. This 'Design–Actuality' gap represents the difference between where an organization currently is, and where it should be in the future if it is to be successful in implementing new systems. These four elements are the process, people, structure and technology. In the context of new information systems in both the developed and developing worlds, Rezaeian and Wynn [36] have taken technology, people, and processes as three dimensions of change for analyzing such projects in Iran. Structure was considered as a function of major process change, and thus was omitted from their analysis as a main element. The concept of phases in the introduction of technology related change, and associated strategy is also evident in some of the literature. For example, Rezaeian and Wynn [36] identify three distinct phases—Pre-Implementation, Implementation, and Post-Implementation; while Zhu, Kraemer, and Xu [37] similarly suggest that digital innovation consists of three phases—initiation, adoption and routinisation. Similarly, Pichlak [38] (p. 476) sees the innovation process in organisations as "a sequence of stages, progressing from initiation through adoption decision to implementation".

*2.5. Conceptual Framework*

The review of the extant literature indicates a dearth of strategy frameworks to guide the development and implementation of e-business in a developing world environment, especially for small businesses. Nevertheless, there are relevant models and concepts in the current literature, and the conceptual framework for the study builds upon some of them. It takes the three dimensions of process change, people skills development and technology deployment, and assesses these across three phases of e-business strategy activity in each of the case studies. These three phases are development, implementation and review (Figure 1).

The process change dimension encompasses a range of activities aimed at making an organization's workflow more effective, more efficient and more capable of adapting to the changes required by the deployment of new technology. It is based on the process management discipline, which attempts to reduce human error and miscommunication and focus stakeholders on the requirements of their roles. Elements of this dimension are documentation, adaptability and accessibility. The people skills dimension encompasses a range of factors at different levels within the company. At the executive level, it concerns managers' leadership, commitment and support and the flexibility of management systems; at the operational level, employees' uptake of new skills and availability of training programmes, resistance to change, lack of awareness and conceptual understanding are all elements of change. The technology deployment dimension focuses on the e-business systems and technologies that need to be implemented. Software, hardware (including tablets and smartphones), communications infrastructure, internet access, power supply and the necessary standards and protocols are all elements of the technology dimension. This outline framework, derived from existing literature, was used in conjunction with the case study profiles discussed below to develop activities and expected outcomes in each phase of the model.

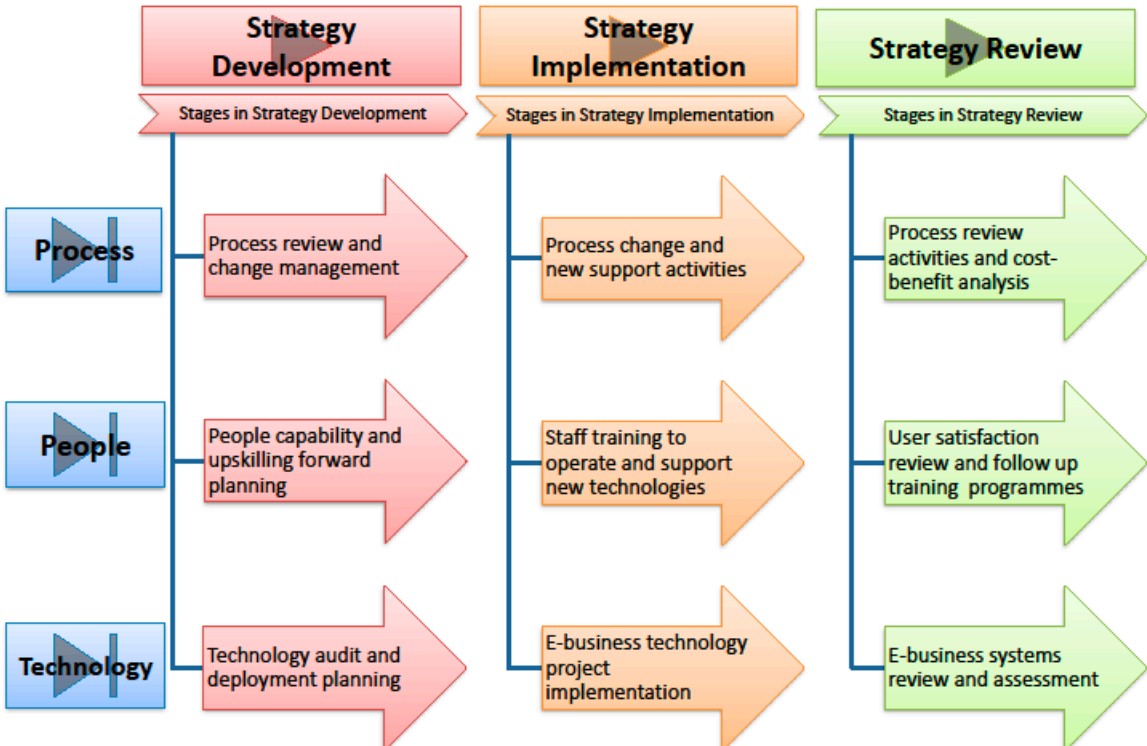

**Figure 1.** Conceptual framework for e-business strategy development, implementation and review.

## 3. Research Method

The initial research that investigated the e-business status of six Nigerian SBEs adopted a qualitative inductive approach, which has been reported elsewhere [10,39]. As noted by Yin [40], a case study strategy is well suited for research that attempts to answer "how" and "why" questions, and Noor [41] also underlined the value of case studies in studying complex real-life phenomena in depth. The six case study companies were sent a detailed questionnaire, which was filled in by the business owners and relevant management staff. This was followed up with semi-structured interviews. For coding and thematic analysis, the six steps identified by Gray [42] were followed. Themes and sub-themes were identified from a rigorous process of reading and re-reading of the extended text generated from interview transcriptions. By applying open, axial and selective coding techniques [43], themes relevant to the phenomenon being studied emerged. NVivo was used as the qualitative analysis tool to code the data.

Based on an analysis of existing literature, Tong, Sainsbury and Craig [44] developed a 32-item checklist for conducting interviews, sub-divided into three main domains: research team and reflexivity, study design and analysis and findings. The research reported here emanates from a new enaction of the third phase of this cycle, notably in the derivation of new themes following the initial case study profiling briefly summarized below. Table 1 provides an example of how this checklist can be applied. An inductive approach was further applied to build upon emergent themes [45] and develop an e-business strategy framework based on the findings from that initial research. As such, only a summary overview of the six case study companies and their use of e-business is provided here. The focus of this article is the development of the framework for e-business strategy development, implementation and review, not on a detailed analysis of the case studies.

**Table 1.** Checklist for interview analysis and findings domain in this study. Adapted from ref. [44].

| Item | Item Description | Assessment |
|---|---|---|
| 24 | Number of data coders. How many data coders coded the data? | One |
| 25 | Description of the coding tree: Did authors provide a description of the coding tree? | No |
| 26 | Derivation of themes: Were themes identified in advance or derived from the data? | Derived from data and conceptual framework |
| 27 | Software: What software, if applicable, was used to manage the data? | NVivo |
| 28 | Participant checking: Did participants provide feedback on the findings? | Yes |
| 29 | Quotations presented: Were participant quotations presented to illustrate the themes/findings? Was each quotation identified? e.g., participant number | Yes |
| 30 | Data and findings consistent: Was there consistency between the data presented and the findings? | Yes |
| 31 | Clarity of major themes: Were major themes clearly presented in the findings? | Yes |
| 32 | Clarity of minor themes: Is there a description of diverse cases or discussion of minor themes? | Both |

The development of the framework was based on perspectives provided in the initial semi-structured questionnaires and interviews, and validated through follow-up interviews reported on in Section 5 below. Interpretivism was adopted as the philosophical approach. This paradigm recognises that the world can be influenced by various actors and factors, and within the context of Nigerian SBEs, using an interpretivist lens provided a distinct viewpoint through which a range of factors affecting e-business strategy could be studied. Qualitative case studies tend to support higher-value findings related to understanding and explaining context, relevance and causality of the phenomena [46]. The six SBEs (distilled from an initial survey of 47 SBEs) are from a variety of industry sectors in Nigeria. Figure 2 indicates their location in Nigeria and the geo-political zones in which they operate. A brief profile of the companies is given below, with pseudonyms being used for company names.

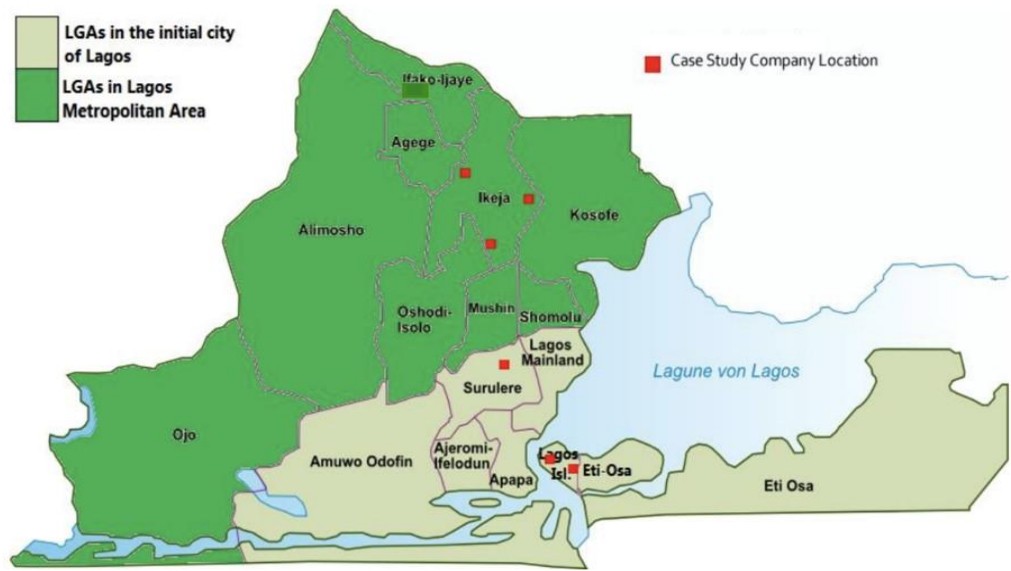

**Figure 2.** Location of the six case study companies.

*ABC Laundries*, based in Lagos, Nigeria, is a budget laundry and dry cleaning business, founded in 2010. The company's main operations office is in Surulere within the Lagos University Teaching Hospital, from where the company offers its laundry services to students and staff at the hospital. In addition, it offers pickup and delivery services to companies, corporate services, and guesthouses across Lagos State. Annual revenues are in excess of 6 million Naira (USD 14,000) per annum, and the staff headcount is seven. (Average annual wages are less than USD 1400 for these staff, significantly lower than in the developed world). The business plan is to grow market share, increase the customer base, and drive up turnover.

The first investment in e-business systems came in 2013, when the company developed a bespoke web portal (using PHP and the MYSQL database) for its sales, marketing and associated financial management processes, replacing a combination of spreadsheets, open source products and paper-based systems. The new system allowed order capture and processing via the web at both the company's locations, provided an order-tracking facility for staff and customers and provided financial management information. In addition, customer and contact details were stored in one database. The system is integrated with email servers and SMS gateways, thus facilitating SMS notifications and emails to be sent to customers. Table 2 provides an overview of systems currently deployed.

**Table 2.** IT/e-business systems at ABC Laundries.

| IT/E-Business System | Process Areas |
| --- | --- |
| Web Portal | All Process Areas |
| MTN Cloud IVR | Customer Services; Sales and Marketing |
| Biometric system | HR and Admin |
| MS Excel | Customer Services; Sales and Marketing; HR and Admin |
| Internet Banking | Financial Management; HR and Admin |
| Bulk SMS Portal | Customer Services; Sales and Marketing |
| Job Boards | HR and Admin |
| POS Terminals | Finance Management |
| Email Software-as-a-Service | Customer Services; Sales and Marketing |
| Social Media | Sales and Marketing |
| Website | Sales and Marketing |

*GPY Properties* was founded in 2012 as a property development and marketing company, when it was hived off from a larger property management enterprise. The company's mission is to provide innovative, high quality and affordable homes to help address the country's housing shortage. Annual revenues are around 35 million Naira (USD 84,000).

The company's e-business systems include its website, where its current property development projects are showcased and its property portfolio is marketed, and a cloud based Customer Relationship Management (CRM) application used for storage and analysis of client information. The company also uses Facebook and other property websites for advertisements and promotions. Excel spreadsheets are used for invoicing and related accounting activities, but the Wave Accounting and Xero Accounting packages are being assessed as possible replacements. The company has just three full time staff but twenty contract staff. By using a range of systems (Table 3), most of the company's daily business activities are now automated, including internal communication, customer engagement and product marketing.

*KDY Energy* was founded in 2012, when the current owner returned from studying electronics in the UK and identified an opportunity for alternative energy solutions. Using solar technologies, the company aims to supply both residential and commercial customers with cost effective solutions. The company has two full-time staff and eight temporary staff, and annual revenues are over 100 million Naira (USD 240,000).

**Table 3.** IT/e-business systems at GPY Properties.

| IT/E-Business System | Process Area |
|---|---|
| CRM | Customer Services |
| | Property Sales and Marketing |
| Website | Property Sales and Marketing |
| | Customer Service |
| | Customer Services |
| MS Excel | Financial Management |
| | Payroll and HR Management |
| Bulk SMS Portal | Customer Services |
| Email Campaign Manager | Customer Services |
| | Property Sales and Marketing |
| Social Media (Facebook, Twitter) | Property Sales and Marketing |
| | Customer Services |
| Internet Banking | Financial Management |
| | Payroll and HR Management |
| Job boards | Payroll and HR Management |
| WhatsApp | Customer Services |
| QuickBooks | Financial Management |
| Online Property Aggregators | Property Sales and Marketing |

The company's website was set up in 2014 to market its products and services to potential customers as part of its business strategy to expand its customer base, particularly in the commercial sector. Excel spreadsheets are used for day to day financial recording, for expenses and other outlays, but for its quarterly accounting the company uses QuickBooks via the cloud. (Table 4). The company used supplier websites for its procurement process, but currently the company makes only minimal use of social media for its own marketing. There is no online order capture capability at present, but the owner is aware of the potential of e-business and is keen to incorporate an enhancement of its e-business operations in future business plans.

**Table 4.** IT/e-business systems at KDY Energy.

| IT/E-Business System | Process Areas |
|---|---|
| Website | Sales and Marketing |
| QuickBooks | Financial Management |
| MS Excel | Procurement and Logistics Management; Financial Management; Payroll and HR Management |
| Email (Webmail) | Customer Services; Sales and Marketing; Payroll and HR Management; Procurement and Logistics Management |
| Internet Banking | Financial Management; Payroll and HR Management |
| Logistics Tracking | Procurement and Logistics Management |
| Supplier E-commerce Sites | Procurement and Logistics Management; Financial Management |
| Inverter Monitoring | Installation and Repair |
| WhatsApp | Customer Services; Sales and Marketing |
| Social Media (Facebook and Instagram) | Sales and Marketing |
| Job Boards | Payroll and HR Management |

*NUGX Consulting* was founded in 2007 by a formal civil servant and operates in Lagos, the country's capital. The company's core business is training and consultancy, with annual revenues of over 10 million Naira (USD 24,000), with customers being mainly in Nigeria but also in other parts of West Africa. The company has four full-time staff and three contractors, providing standard or tailored in-house courses to customers mainly in

the financial sector. Training and consultancy themes include customer service, agency marketing, life insurance and general management.

The company's website provides information on future training courses and customers can make bookings via the website but the financial transactions are processed off-line. The company uses a bespoke customer relationship management (CRM) application to store customer data and maintain contact with current and past customers and course attendees. The company distributes a newsletter electronically but also uses hard copy fliers, distributed via dispatch riders, for the less IT-enabled customers. The company uses a range of other IT tools as shown in Table 5.

**Table 5.** IT/e-business systems at NUGX Consulting.

| IT/E-Business System | Core Process Areas |
|---|---|
| Website | Sales and Marketing |
| eFront (Learning Management System) | Curriculum and Training |
| CRM | Sales and Marketing |
| | Customer Service |
| Internet Banking | Financial Management |
| | Payroll and HR |
| MS Excel | Payroll and HR |
| | Financial Management |
| | Curriculum and Training |
| Ms PowerPoint/Word | Curriculum and Training |
| | Sales and Marketing |
| | Customer Service |
| Email | Sales and Marketing |
| | Payroll and HR |
| | Financial Management |
| Job Board | Payroll and HR |
| Facebook/LinkedIn | Sales and Marketing |
| Paystack Payment Pages | Sales and Marketing |

*HGB Stores* is a retail business founded in 2015 and has just five staff. The company designs, makes and sells children's clothing and was originally set up in the founder's kitchen. Initially, the business started with her collecting imported materials and reselling to friends and family, but the company now has an average of 20 orders per day with an annual turnover of 19 million Naira (USD 45,000). The company has adopted e-business technologies as needed and as budgetary constraints allowed. E-commerce sites and marketplaces are accessed to sell the company's products, and a number of tools and technologies are used as shown in Table 6.

As the company grew and its client base and sales revenue began to increase, Microsoft Excel was increasingly used for data collection, management, analysis and reporting, while cloud-based systems such as SMS live and Mailchimp are used for customer communication. Although a number of systems in the firm still make use of Microsoft Office systems, the company made considerable use of social media platforms to generate sales (via social media advertising) and made use of easy to deploy solutions to accept payment from customers online, enable delivery tracking and provide customer updates via SMS. The company accepts online payments through Paystack payment pages, which allowed Nigerians in the diaspora to easily purchase gifts for friends and family in Nigeria from the company's Instagram Store. Android tablet computers provide staff with reliable battery-based machines for most of the days' activities, as power supply is a significant concern.

*OMO Legal* was founded in 1971 and now has 15 staff providing a range of legal services to business and private individuals. It has three branches across Nigeria and its practice areas are litigation and arbitration, regulatory enforcement, acquisition and

takeovers, and intellectual property. Turnover is circa 36 million Naira (USD 86,000). The company's first computer was purchased in the 1990s, and there has since been a gradual transition from manual operations to make use of Microsoft Office packages. In 2010, the return of the (now) principal partner to Nigeria, after a period working in the UK, was the catalyst for the introduction of the website, a case management system, Sage Accounting and Dropbox for Business.

**Table 6.** IT/e-business systems at HGB Stores.

| IT/E-Business System | Core Processes |
| --- | --- |
| Instagram | Customer Services |
| | Sales and Marketing |
| MS Access | Customer Service |
| | Customer Services |
| | Financial Management |
| MS Excel | Inventory Management |
| | Logistics and Delivery Management |
| | Sales and Marketing |
| WhatsApp | Customer Services |
| | Sales and Marketing |
| | Customer Services |
| Email | Sales and Marketing |
| | Procurement Management |
| E-commerce Sites and Marketplaces | Procurement Management |
| Internet Banking | Financial Management |
| Third-party Tracking API | Logistics and Delivery Management |
| Canva (Graphics Design) | Sales and Marketing |
| | Financial Management |
| Paystack Payment Pages | Sales and Marketing |
| | Customer Services |
| Bulk SMS (Transactional and Promotional) | Sales and Marketing |
| Buffer (Social Media Management) | Sales and Marketing |

This gave the company a point of difference compared with many of its competitors, leading to its expansion to other locations. New staff have brought in knowledge of other tools and technologies, based on previous experience. E-business transactions can now be processed via the company's website, and a range of systems is used by company staff (Table 7). The firm's case management system operates as both a record-keeping system for cases as well as a customer relationship management system to store and report customer data. A self-hosted email system is used for external communication with clients, while Sage Accounting is used for managing the company's accounts. Dropbox for Business allows operational documents to be viewed and exchanged across multiple locations. It is planned to integrate the Sage accounting and case management systems, thus allowing for easier billing for customers.

In summary, the six case studies indicated a clear transition to using e-business systems over the past 5–10 years, with a wide range of systems and technologies being deployed. The approach has been largely tactical and practical, rather that holistic and top-down, although a loose alignment with business objectives was evident in most cases. However, in all six companies there was an absence of any framework for e-business strategy development and implementation, with companies normally adopting an ad-hoc approach, with no clearly defined roadmap. Building upon the lessons learnt in these case studies, the following section puts forward such a framework.

**Table 7.** IT/e-business systems at OMO Legal.

| IT/E-Business System | Core Processes |
| --- | --- |
| Website | Business Development |
| | Customer and Case Management |
| MS Excel | Business Development |
| | Admin and HR Management |
| | Financial Management |
| | Research and Paralegal |
| Case Management | Customer and Case Management |
| Email | Customer and Case Management |
| | Financial Management |
| Sage Accounting | Business Development |
| | Financial Management |
| Internet Banking | Financial Management |
| | Admin and HR Management |
| Facebook | Customer and Case Management |
| | Business Development |
| DropBox | Customer and Case Management |
| | Research and Paralegal |
| MS Word | Research and Paralegal |
| | Admin and HR Management |

## 4. The E-Business Strategy Framework

The framework outlined in this section builds upon the conceptual framework, and comprises three phases—strategy development, strategy implementation and strategy review. Each phase contains a number of linked stages (for which the acronyms SAPP-STEER-SCOR can be used), which each have a range of associated activities for people, process and technology change dimensions. More detail on all stage activities is available online [10].

### 4.1. Strategy Development

Strategy development consists of four stages: Specify goals and objectives, Analyse current situation, Prioritise action areas and Plan strategy implementation (SAPP). The people, process and technology activities to be considered in each of these stages are shown in Figure 3, and some illustrative detail of the four stages is provided below.

### 4.1.1. Specify Goals and Objectives

Small businesses vary in size, structure and focus, and the reasons for adopting e-business will vary from company to company. In the case study companies, the motivation for adopting e-business included increasing efficiencies, growing market share, and outperforming the competition. The primary goals and objectives of the e-business deployment need to be made clear, and be aligned with the company's overall business goals. Mapping the core business processes provides a framework for identifying underperforming areas of the business where new e-business systems or technologies may be implemented as at ABC Laundries (Figure 4). It also helps identify constraints in certain process areas that may hamper e-business initiatives as at KDY Energy (Table 8).

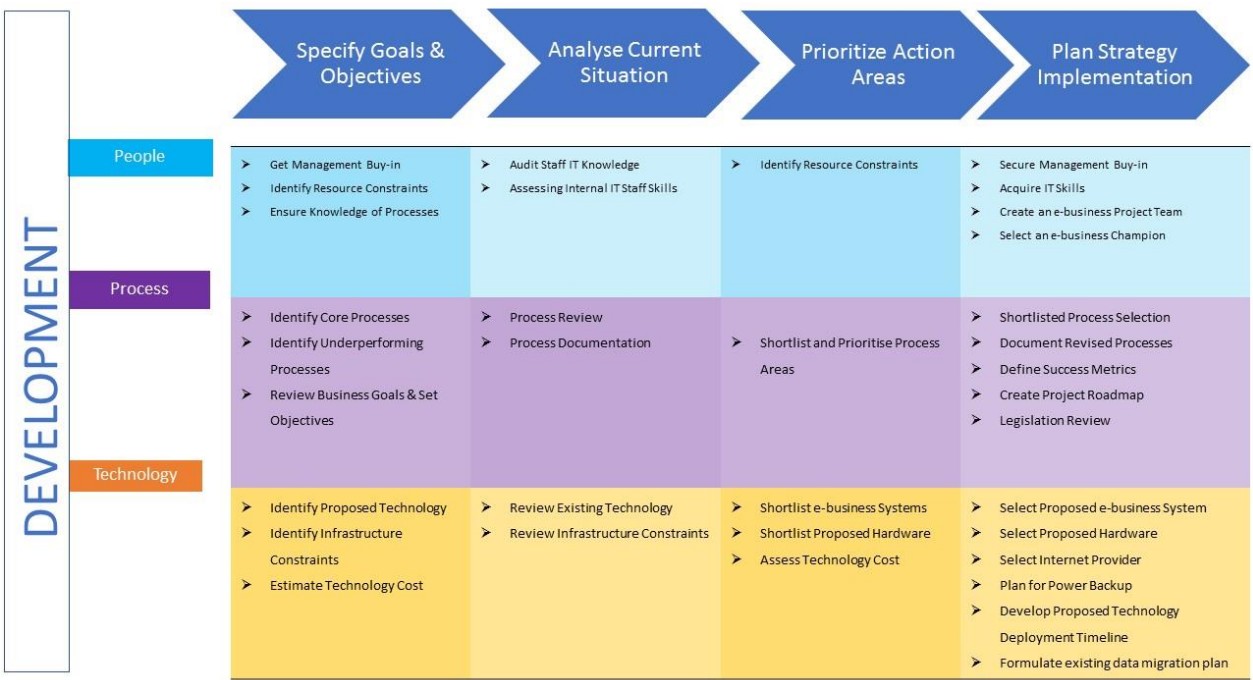

**Figure 3.** E-business strategy development framework [10] (p. 225).

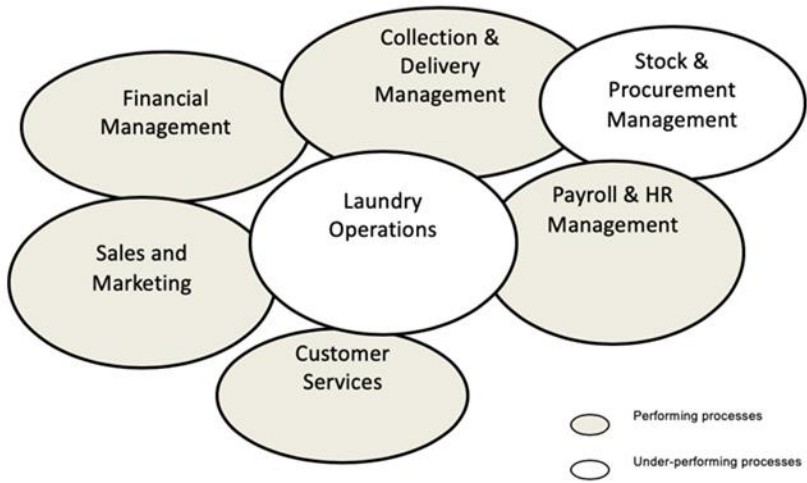

**Figure 4.** Underperforming processes at ABC Laundries.

**Table 8.** Process constraints KDY Energy.

| Core Process | Sub-Processes | Constraints |
|---|---|---|
| Financial Management | Company Accounting | Payments are usually made to different bank accounts. |
| | Invoicing | Several expenses are cash-based and difficult to track. |
| | Daily Expense Tracking | Not all expenses have receipts. |
| Installation and Repair | Installation | This process involves the engineer visiting the site. |
| | Repair | This process largely involves several third parties—suppliers, clients and logistics provider. |
| | | Appointments are subject to availability of engineer, equipment and the client location. |
| Sales and Marketing | Order Management | This largely involves working with several large organisations with limited IT drive. |
| | Lead Generation | |
| Logistics and Procurement | Procurement | Involves working with some local suppliers who use limited IT systems |
| | Goods Tracking | Logistics companies used are generally sole traders |
| | Inventory | |
| Customer Services | Problem Resolution | N/A |
| | Messaging | |
| Payroll and HR Management | Payroll | N/A |
| | Recruitment | |

### 4.1.2. Analyse Current Situation

This stage focuses on helping businesses analyse their existing technology, process and people capabilities to determine what changes and investment must be made in order to implement e-business successfully. This will include, in the people dimension, for example, a skills gap analysis assessment of current staff (Table 9) Similarly, in the technology dimension, an audit of existing systems and technologies should be undertaken (Table 10). The assessment should be thorough enough to avoid replicating systems and planning based on outdated processes and procedures.

**Table 9.** Skills gap analysis template from ABC Laundries.

| Staff Name: Role: Developer/IT Manager | | | | |
|---|---|---|---|---|
| Competency | Behavioural Description | Lacking | Improve | Competent |
| MYSQL | Advanced knowledge of MYSQL database and server | | ✓ | |
| Web programming (PHP, ASP.NET) | Good knowledge and experience of web application programming | | | ✓ |
| Web design | Knowledge of HTML and CSS | | | ✓ |
| API | Advanced knowledge of API development, securing API and integration of third-party API | | ✓ | |
| Apache webserver | Understanding of Apache web server and experience with debugging | | ✓ | |
| Experience with Digital Ocean | Experience working with Digital Ocean as a service provider | ✓ | | |
| Networking | Advanced understanding of networking, routing and DNS | ✓ | | |
| JavaScript | Knowledge and experience working with JavaScript particularly using jQuery | | ✓ | |
| WordPress | Knowledge and experience working with WordPress to develop and manage website | | | ✓ |
| Ubuntu Linux | Comfortable working with Linux servers; this includes deploying web applications, managing them and basic system administration | | ✓ | |
| Software Testing | Experience using automated testing frameworks | | ✓ | |
| Firewall Configuration | Experience working with Iptables on Linux web servers | | | |
| Git | Advanced Knowledge of Git using GitHub and Bitbucket | | ✓ | |

**Table 10.** Technology review at GPY Properties.

| IT/E-Business System | Status | Comments |
|---|---|---|
| CRM | OK | |
| Website | Needs adjustment | Difficult to update. Have to rely on the developer. |
| Bulk SMS Portal | External | |
| Email Campaign Manager | External | |
| Social Media (Facebook, Twitter) | OK | |
| Internet Banking | External | Difficult to export bank transactions data into the accounting system. |
| Job boards | External | |
| WhatsApp | OK | |
| QuickBooks | External | |
| Online Property Aggregators | External | |

### 4.1.3. Prioritise Action Areas

This stage focusses on prioritising action areas for the e-business initiative, identifying business processes where e-business systems could be implemented, shortlisting systems and exploring some cost implications. The firm should be answering questions such as: What systems do we want to implement now (and later)? What do we need for the e-business systems to operate effectively? Will the new processes differ from the existing processes?

#### 4.1.4. Plan Strategy Implementation

This final stage of the strategy development phase is concerned with finalising all pre-adoption activities, getting everything that needs to be in place ready for implementation of the e-business strategy and putting together a realistic plan for strategy implementation. This will include a project road map as at NUGX Consulting (Figure 5).

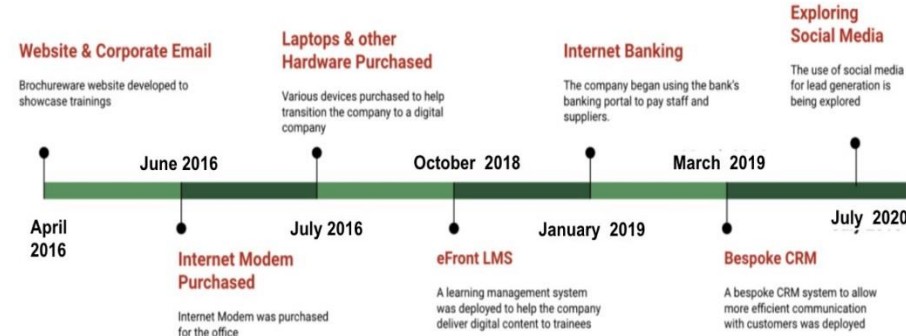

**Figure 5.** Project roadmap for NUGX Consulting.

#### 4.2. Strategy Implementation

In the strategy implementation phase, the selected e-business systems are developed, purchased and deployed. Given Nigeria's challenges with power and slow internet connections speeds, power backup and internet provision will need to be purchased and deployed in the organisation as necessary. The objective is to guide the organisation through a stage-based implementation process, ensuring there is value derived and the success metrics can be achieved. This phase, shown in Figure 6, consists of five stages: Startup, Trial, Expand, Embed and Review (STEER).

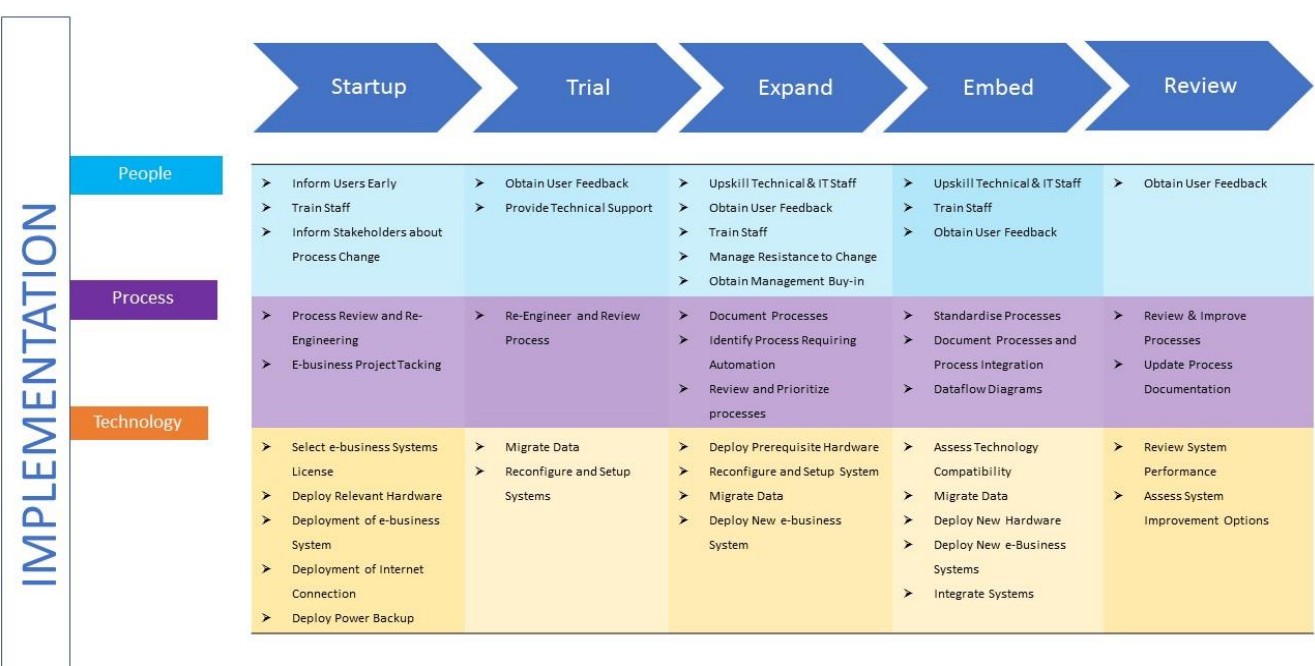

**Figure 6.** E-business strategy implementation framework [10] (p. 243).

### 4.2.1. Startup

This stage initialises the strategy implementation phase of the framework and includes activities such as procurement of systems, process reviews, and power backup installation. The pre-requisites for new technology implementation are put in place and training for IT staff and general users of the systems is carried out. Staff members in the organisation should be briefed about the e-business project and how this intends to help the company achieve its core goals. A simple project tracking process should be embarked upon, as at ABC Laundries (Table 11).

**Table 11.** Project tracking plan at ABC Laundries, February 2018.

| Project Item | Deadline | Status |
|---|---|---|
| Development and Testing of Invoicing Module | March 2017 | Completed |
| Deployment of Invoicing Module | April 2017 | Completed |
| POS and Thermal Printer Deployment | April 2017 | Completed |
| Staff Training on Inventory Module | April 2017 | Completed |
| Development and Testing of Laundry Status Tracking | May 2018 | Completed |
| Deployment of Laundry Status Tracking | June 2018 | Completed |
| Integration of SMS gateway | January 2019 | Completed |
| Development of Accounts Module | August 2017 | Completed |
| Website Development | July 2018 | Completed |
| Deployment of Accounts Module | September 2017 | Completed |
| Development of Reports Module | December 2018 | Completed |
| Deployment of Biometric Attendance monitor | December 2017 | Completed |
| Development of Inventory Module | October 2017 | Completed |
| Development of Customer Profile Module | January 2018 | Completed |
| Development of Automatic email and SMS reminders | December 2017 | Completed |
| Deployment of Inventory Module | March 2018 | In progress |
| Development of Delivery Schedule | June 2018 | Completed |
| Development of Expense Tacking | January 2018 | In progress |
| Deployment of Customer Profile Module | April 2018 | Not started |
| Deployment of Customer Profile Module | June 2018 | Not started |
| Deployment of Delivery Schedule | August 2018 | Not started |
| Deployment of Expense Tacking | July 2019 | Not started |

### 4.2.2. Trial

This stage is concerned with a more detailed pilot implementation to gather feedback and understand how the implementation will affect the organisation. This is based on the priorities and roadmap drawn up in in the strategy development phase. By piloting in one or two process areas, benefits can be monitored, and business impact and costs closely managed.

### 4.2.3. Expand

This stage is concerned with evaluating the results of the trial stage, learning from it, and deciding on the possible expansion of implementation to other process areas. This might mean, for example, moving from internal process deployment to externally facing processes, typically sales and marketing. In the case studies, the trial and expand stages stood out as contributing to successful project implementation.

### 4.2.4. Embed

Here, the focus is on embedding various e-business systems across a number of business processes and integrating systems, ensuring data can be shared between various systems and business processes. This should drive home the benefits of improved performance and efficiency in the organisation.

### 4.2.5. Review Progress

In this stage, the focus is on reviewing, improving and optimising existing systems that have been implemented in order to generate increased value for the organisation. The company should gather feedback from the various stakeholders in the organisation, review, reflect on it and identify areas for improvement.

### 4.3. Strategy Review

This phase of the framework is concerned with tracking what has gone well and what has not and learning lessons as appropriate. The status of current e-business systems is assessed, cost-benefit analysis is conducted, and the objectives and goals of the e-business initiative set out at the Strategy Development phase are reviewed. As the overall business strategy evolves, so will the e-business goals and objectives need to be reviewed to check overall alignment. There are three stages, as shown in Figure 7: Status assessment, COst-benefit analysis and Review Goals and Objectives (SCOR).

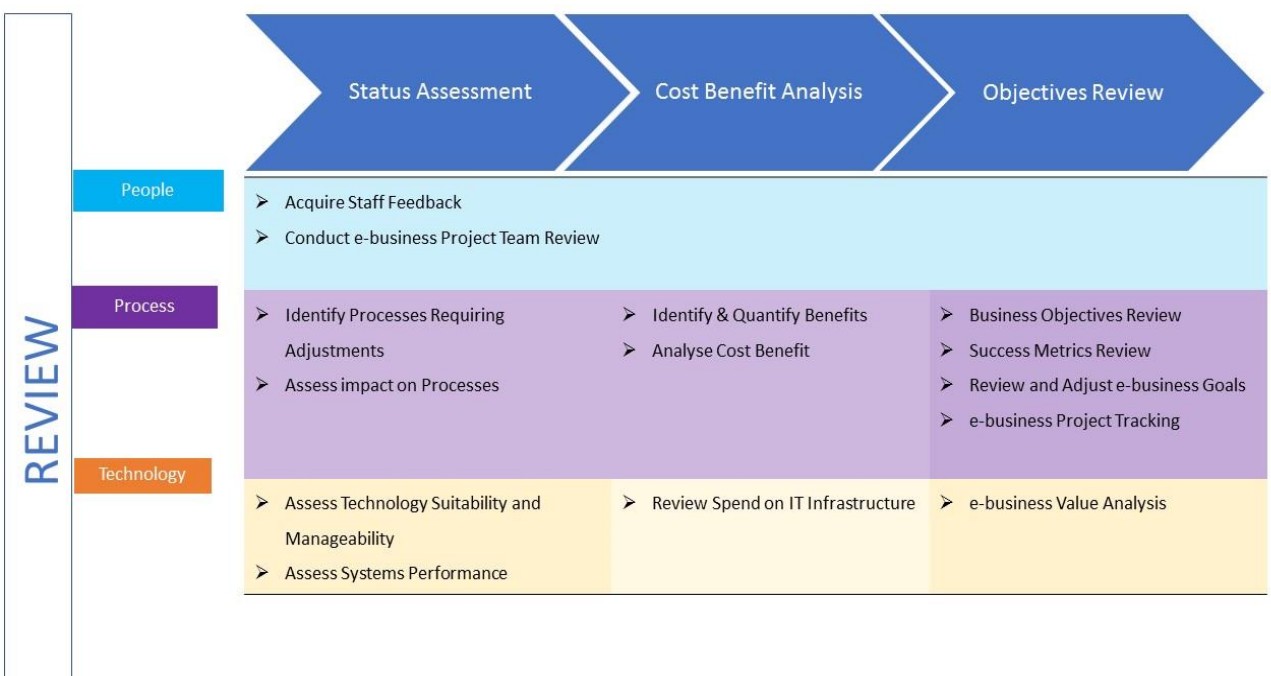

**Figure 7.** E-business Review framework [10] (p. 261).

#### 4.3.1. Status Assessment

This stage is concerned with assessing the current status of the e-business systems implemented. This assessment should be all-encompassing, covering key areas such as assessment of technologies used, assessing the capacity of technology resources, gathering user feedback and identifying the impact on processes.

#### 4.3.2. Cost-Benefit Analysis

Companies will need to compare what has been achieved with the success metrics defined in the Strategy Development phase, and ascertain if projects and initiatives have achieved expected benefits. Activities will include identifying and quantifying benefits and reviewing spend on technology.

#### 4.3.3. Review Goals and Objectives

Goals and objectives set out in the Strategy Development phase are reviewed and reassessed in light of any developments in the overall business strategy. It is also an opportunity for the project team to review suggestions for improvement in their set up, team composition and operation.

## 5. Framework Validation and Guidelines

### 5.1. Framework Validation

The SAPP-STEER-SCOR e-business strategy framework was validated through semi-structured interviews with SBE owners and IT professionals who were familiar with e-business and SBEs in Nigeria. Twelve participants were recruited to participate, six being the owners (or principal partner) of the case study companies, and the other six being Business Analysts or Project Managers with at least ten years' relevant experience.

In the discussion below, the following codes are used:

P1: Principal Partner OMO Legal; P2: Owner of HGB Stores; P3: Owner of ABC Laundries; P4: Owner of KDY Energy; P5: Owner of GPY Properties; P6: Owner of NUGX Consulting.
P7: Business Analyst1; P8: Business Analyst2; P9: Business Analyst3.
P10: Project Manager1; P11: Project Manager2; P12: Project Manager3.

The validation interviews were loosely structured around five main questions relating to the framework, but generally focused on the applicability of the framework within the Nigerian SBE sector and its relative strengths and weaknesses.

The six company owners/partners were asked to identify what phase and stage of the framework their companies were at. The other six participants were asked to identify an SBE in Nigeria with which they had worked and use that as a basis to assess the framework. All the participants were able to do this, and in most cases, the companies were in the Strategy Implementation phase at or between the Trial and Expand stages. P1, for example, noted, "I think we are still in the implementation phase. We're still doing a bit of trialling, leading up to further expansion . . . Some of our staff are still trying to get a grip of some of the systems, which is going to take time".

When asked if they could identify with the people, process and technology elements in the framework, the participants indicated that the framework was comprehensive, relevant and reflected industry practices within the context of Nigerian SBEs. They also commented that they could associate with the activities in the various stages. Ten of the participants were able to briefly discuss how each of the people, process and technology elements related to their company in the stage they were currently at. P2, however, whose company, HGB Stores, was at the Trial stage in the Implementation phase, noted "because we are still at the Trial stage, emphasis has been placed more on technology and the people, because we need to let our staff know what it's all about, why we're introducing such technology and e-business systems to the organisation, the purpose and the advantages it brings".

The participants commented that the framework appeared to encompass all the activities, stages and phases relevant to SBEs in Nigeria. P3, for example, observed, "I believe that it has the relevant components required for e-business deployment for small businesses in Nigeria". P7, however, suggested a review of relevant legislation would be of value when developing strategy. To address this a "legislation review" has been added as a process element in the final stage of the Strategy Development phase (Figure 3).

The e-business framework can be used alongside other methodologies, and this was alluded to by some interviewees. P8, for example, observed, "in project management we have something we call feasibility and that's just a stage of the project where you look at almost all the same things I can see in the Development phase (of your framework). So, I think all the things I've seen on here are fine and representative of an e-business project. Although they are sometimes referred to in different terminologies from what I am used to, this is fine, as terminologies change from organisation to organisation. From my experience, we usually have four stages of a project, but having gone through all this again, all the stages in this framework are adequate. It is what's under each of the stages that makes it rich. So, I think the stages are fine."

When asked specifically if the framework would be considered useful for SBEs in Nigeria to develop and implement e-business strategy, the responses were generally positive, with several participants commenting on the frameworks' robustness, clarity, simplicity and that it provided a clear roadmap. P1 was very supportive, saying:

It's a fantastic structure here. I think this is really simple and it's just self-explanatory. And it's like a guiding process for everyone from planning strategy implementation, to start up, trial and other stages. I think it's great. I think this will really help and I would definitely recommend this to anyone running a small business in Nigeria. This is really good.

P9 noted:

Considering the purpose of this framework and the way you have put it together, I think it's really concise and very straightforward. So, I believe it's something that's achievable for small business owners in Nigeria, to be able to adapt this particular framework, to develop e-business in their businesses. So, it's really neat. For the first time I saw something like this, I just felt, okay a common businessman in Nigeria with less than 50 employees, should be able to adapt to this because it's quite straightforward.

P4 highlighted the value of the framework as a day-to-day guide. He commented:

I think it is very useful. I think it breaks it down into little small chunks, so that . . . as you are going through one stage, you can pick individual elements within the people process and technology aspects to address. It is a very helpful guide. Many times, in a big organisation, things are already set up. I think more than anywhere else, guides like this are very important and useful in small businesses where you are trying to put structure in place.

However, two participants (P10 and P11) indicated that given the detailed nature of the framework, it might be difficult for business owners, without technological knowledge and adequate technical support, to use the framework for the purposes of developing and implementing e-business strategy. One participant suggested that the framework might be better targeted at e-business implementers with technical knowledge rather than everyday SBE owners, while the other suggested a trimmed-down version (without so many activities), together with a detailed guide, might better serve business owners. On the other hand, four of the SBE owners, without IT backgrounds, indicated that they were able to relate to the framework and saw how it could be used to help develop strategy and implement e-business in their organisations.

Concerning the usefulness of the framework in other environments and with SMEs, the interviewees were generally positive. One of the IT project managers who had spent years working in Ghana commented on the similarity between Nigeria and Ghana, but advised that, to use the framework in Ghana, some adjustments might need to be made in the people and technology elements to adapt the model to the realities of local technology availability and people skills.

P11 indicated that they found the framework comprehensive and yet easy to understand. Several comments were also made about the framework's robustness, in that it did not simply focus on the implementation phase, but provided detailed activities in the development and review phases, thus allowing for continuous improvement of the business. P5 noted:

"One of the core strengths of the framework is that it shows you exactly what you need to do", and P10 commented that "by providing clear activities in each of the people, process and technology dimensions in all the phases and stages of the framework, the framework is clear and easy to follow".

P3 saw particular value in the framework in that it builds upon "best practice from business and from technology" and added, "the combination of people, process and technology elements was a key strength".

P7 suggested that the number of stages and phases in the framework might result in a slow implementation of e-business in an organisation. He gave the example of the final stage of the Development phase, in which securing management buy-in might result in a

delay in the implementation. At the same time, however, he conceded that the governance structure put in place by the framework is an advantage for bigger projects.

In summary, the feedback from the 12 participants suggested the framework was:

- Realistic and relevant to the SBE environment in Nigeria.
- Comprehensive in that it encompasses multiple dimensions of change.
- Easy to use and understand, with some qualifications regarding the need for knowledge of technologies involved.
- Adaptable to different company sizes and business environments.
- Very useful as a roadmap that could overlay, and integrate with, other methodologies that may be in use.

*5.2. Guidelines for Framework Deployment*

The framework is designed to be simple to use and self-guided, but a small business owner or manager will likely require the expertise of an IT manager or staff member to fully implement all the activities within the framework.

Following the validation interviews, and building upon interviewee recommendations and comments, a sequence of steps for using the framework is suggested below:

- *Establish where you are in the Development–Implementation–Review cycle*: Many companies will already have embarked upon some form of e-business project. Identify what phase within the framework the company is at—Development, Implementation or Review?
- *Identify which stage or stages match your current situation:* By identifying the current stage, the user can focus on the process, people and technology activities required to move forward effectively.
- *Review and reaffirm business goals and e-business objectives*: Take a step back, review and reaffirm business goals and e-business objectives. Have a look at the process activities in the Review Goals and Objectives stage of the Strategy Review phase. Clarity of the business objectives is essential for the successful implementation of an e-business strategy in a small business.
- *Familiarise management and staff with the process, people and technology elements of the framework*: The framework emphasises an equal balance of process, people and technology change dimensions that are necessary for e-business implementation in an SBE. It is important all involved staff understand and buy-into the fundamental concept of managed multi-dimensional change.
- *Reflect on the previous two stages:* Check that the activities that should have been undertaken in the past two stages are now completed as necessary before moving forward. (This step should be omitted for companies just starting their e-business initiative).
- *Set the course ahead and secure management buy-in:* Review immediate and short-term activities in all three dimensions (process, people and technology), and allocate resources as needed. The owner or managing partners are responsible for making most decisions in SBEs, so their support and familiarity with the framework and its budgetary implications is essential.
- *Plan next stage*: Get management and team structures in place, review resource issues and plan accordingly.

## 6. Conclusions

This article presented an overview of the SAPP-STEER-SCOR framework for e-business strategy development, implementation and review in Nigerian SBEs. The three phases of the framework and their respective stages have been set out and related activities in the process, people and technology change dimensions were identified. The framework supports an iterative approach to strategy development, puts forward a set of logically linked stages for implementation and allows for continuous improvement in the review phase. The majority of the existing frameworks are not geared to assisting small business

owners to pursue e-business initiatives because they are either largely theoretical or developed for implementation by practitioners. In contrast, the framework presented here provides a realistic route map for progressing e-business initiatives, geared mainly to the practicalities of small business management rather than theoretical concepts.

Feedback from the company owners and other practitioners suggested the framework was of value, relevant and usable. The framework builds upon the findings from the six case studies outlined above, but also from other research into Nigerian SBEs [22–24] that focused on influencing factors and barriers to e-business. These studies were of undoubted value in that they identified some of the key issues involved in conducting e-business in Nigeria—IT skills issues, customer attitudes, internet connectivity and other infrastructural elements, regulatory frameworks and the age and culture of the user companies. Here, however, the emphasis is on trying to develop a methodology that will help SBEs recognise the key influencing factors and overcome these barriers in the development and implementation of an appropriate strategy. As such, this research and the resultant strategy framework are more akin to the work of Chen, Ruikar and Carrillo [1], noted above, who attempted to develop a similar framework for companies operating in the construction industry.

This research has its limitations. The resultant framework is nothing sophisticated—it is a simple and practical set of activities that would not be suited to large organisations where operations are more complex and more nuanced methodologies are more appropriate. It has been constructed from findings in the Nigerian SBE sector, and thus its applicability in other business environments needs testing. This is one possible area of future research, and indeed one of the project managers involved in validating the framework suggested it could usefully be translated into French and trialled in francophone African countries. Further research might focus on refining and developing the set of activities and building in clearer resource parameters.

Despite these limitations, the framework is an addition to the existing models of e-business adoption, and in contrast to many of the existing models, it is practical, action-oriented and geared to small business enterprises. It is hoped that it will be of value to such companies operating in developing world environments faced with the issues and challenges involved in e-business strategy formulation and implementation.

**Author Contributions:** Conceptualization, M.W. and O.O.; methodology, M.W. and O.O.; formal analysis, O.O.; investigation, O.O.; writing—original draft preparation, O.O. and M.W.; writing—review and editing, M.W. and O.O.; visualization, M.W. and O.O.; project administration, M.W. and O.O. All authors have read and agreed to the published version of the manuscript.

**Funding:** This research received no external funding.

**Institutional Review Board Statement:** The study was conducted in accordance with the ethics principles and procedures of the University of Gloucestershire, UK, as contained in the publication: *Research Ethics: A Handbook of Principles and Procedures,* available online at: https://www.glos.ac.uk/information/knowledge-base/research-ethics-a-handbook-of-principles-and-procedures/ (accessed on 31 May 2021).

**Informed Consent Statement:** Informed consent was obtained from all subjects involved in the study.

**Data Availability Statement:** A fuller statement of case study data and information is available at source [10].

**Conflicts of Interest:** The authors declare no conflict of interest.

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
