# Peer review of "E-Business Strategy in Developing Countries: A Framework and Checklist for the Small Business Sector"

_sustainability, doi:10.3390/su13137356_

Round 1
Reviewer 1 Report
Please provide clear statements how sustainability is addressed in this study and include it in the text also using references also.
Please include your key findings in the abstract.
Please provide clear 1) introduction/literature review, 2) materials and methods, 3) results and 4) conclusions sections in a logical order and including all elements.
Please provide clear sources for all applied frameworks ect (tables, figures..) including references.
Please provide results and conclusions only based on studied materials and aims of the study. Keep difference to already existing frameworks ect. Present frameworks and aims/questions ect in the beginning.
Please do not use "authors" in limitations and provide suggestions for future research.
Reviewer 2 Report
The abstract is clear and well structured, it presents in a comprehensive way the purpose and added value of the research paper.
It is recommended that no abbreviations are used for the key words- the number should be also limited to 4-5 relevant key words.
The Introduction could be extended by providing more information related to the context of the research.
The topic of E-business in developing countries is very interesting for the research and authors present very well previous initiatives in this field in various developing countries, including Nigeria. The conceptual framework (model for E-business used in the present paper) should be depicted more detailed.
The research objectives should be presented before the case studies. It should be explained why a Table presenting the main IT-E business systems has been developed just for 2 enterprises out of the 6. Instead of this approach, a Table which synthesizes the situation in all 6 enterprises could be developed.
It is very difficult to read Figure 3 (blurry). The same applies for Figure 6 and 7.
The different steps of the analysis of the processes which should lead to e-businesses are presented (apparently) randomly for different enterprises from the 6 selected for the case studies. It should be explained how this selection occurred or a comparative analysis of all enterprises should be provided.
As stated in Table 6 some of the Deadlines are in 2018/2019 and according to the status the measures have not been addressed so far. Some references related to the time framework of the research should be provided.
Some general remarks related to framework validation should be formulated (instead of presenting what each single interviewee mentioned (section 5.1).
In the Conclusion section it should be mentioned, if besides the practical added value brought by the research, any theoretical added value can be mentioned (e.g. the development of the conceptual framework).
Some minor errors related to punctuation should be corrected.
Reviewer 3 Report
The main goal is mentioned in the abstract in a bit obscure way. Try to state the main goal more explicitly. The main goal of the paper id the elaboration and validation of the framework aimed at the development, implementation, and review (or strategic controlling) of e-commerce strategy. If the main goal is focused on strategic framework development, it must be evident that such a research hasn´t been conducted yet and therefore there is a gap. What about the research problems? Which one is the most essential? Lack of strategic thinking of Nigeria´s managers?
It is advisable to set up supporting goal as the „critical assessment of current approaches to the strategy development in the branch of e-commerce. “
Literature part highlights the importance of e-commerce strategies, and it is worth commending that the authors included special chapter dealing with w-commerce strategy in developing countries. Here a recommend to stress more exactly the barriers to smooth implementation of digitalisation and the main barriers to effective penetration of e-commerce in these countries. If the paper deals with innovation adoption it could be illustrative if e-commerce observes any of existing adoption models (e.g., Brand & Huzing´s one)
As for the methodology it is the weakest part of the paper. Necessity to precise goals definition has been already mentioned. Unfortunately, the paper completely misses research questions as well as the responses to them in the discussion part. Contextual interviews, especially their interpretation must be better clarified. Basically, the number of interviews id depending on information saturation (no additional knowledge is excerpted from interviews). The coding and interpretation must be quite persuasive. Which type of coding did the authors use? Open? Axial? Other? The framework elaborated must be appropriately validated. At least by contextual interviews. Fortunately, this was accomplished. Strategy analyst might be curious if popular implementation techniques like balanced Scorecard can be applied in the e-commerce business.
e-commerce adoption is usually perceived as multilevel process:
- Finally, the site may be fully integrated with internal systems.
- The company may include a transaction function for standard products or services.
- The company may add some interactive features by which potential customers get access to information and services tailored to their needs.
- E-commerce may first be used to present the company and its offerings (a “brochureware” site)
Does your framework react to this (or similar) sequential process?
Below you may find some literature resources that can help you to find your bearings in problematics:
- Brand, M.R. and Huizingh, E.K.R.E.(2008). European Journal of Innovation Management, 11(1), pp. 5-24
- Rogers, E.M. (2003). Diffusion of Innovations. 5th edition, Free Press, New York, NY.
- Cooper, J.R. (1998). "A multidimensional approach to the adoption of innovation", Management Decision, 36(8), pp. 493-502.
- Wisdom, J.P., Chor, K.H.B., Hoagwood, K.E. (2013). Innovation Adoption: A Review of Theories and Constructs. Adm. Policy Health, Springer Science+Business Media, New York.
- Pichlak, M. (2016). The innovation adoption process: A multidimensional approach. Journal of Management & Organization, 22(4), pp. 476-49
Reviewer 4 Report
Dear Authors,
Re: Manuscript “E-business strategy in developing countries: a framework and
checklist for the small business sector”
Reviewer’s report:
This paper deals with an interesting topic such as investigating a framework for the development, implementation and review of electronic commerce strategies for small companies in developing countries. The generic case of Nigeria is well justified and the particular cases of companies are equally interesting for countries and companies that are starting out in these aspects. However, for publication I recommend that the authors take into account the following issues:
- No type of discussion is carried out in the conclusions section with the previous theoretical framework, so that the contribution of the study in relation to previous studies is shown more clearly. To do this, it is necessary to create a new discussion section or join it to the discussion section.
- The case studies are not excessive either, although later the interviews offer two participants per case, which makes it difficult to generalize the results and this should have been exposed as a clear limitation. Neither is it offered a script of the interviews carried out, which makes it difficult to read the paper. In this same sense, the authors should have explained the steps and procedures carried out in the application of the methodology, so that the study could gain in validity. For this, I suggest the review of the following literature for the inclusion of these aspects:
- BioMed Central (2017). Qualitative research review guidelines – RATS. London: BioMed Central, https://bit.ly/1WteYxh.
- Tong, A., Sainsbury, P., & Craig, J. (2007). Consolidated criteria for reporting qualitative research (COREQ): a 32-item checklist for interviews and focus groups. International Journal for Quality in Health Care, 19(6), 349-357. https://doi.org/10.1093/intqhc/mzm042
- Yin, R.K. (2018). Case study research and applications: Design and methods. Los Angeles: Sage.
- Dubé, L., & Paré, G. (2003). Rigor in information systems positivist case research: Current practices, trends, and recommendations. MIS Quarterly, 27(4), 597-636. https://doi.org/10.2307/30036550
- Aceituno-Aceituno, P., Casero-Ripolles, A., Escudero-Garzás, J.J., & Bousoño-Calzón, C. (2018). University training on entrepreneurship in communication and journalism business projects. Comunicar, no 57, v. XXVI, 91-100. https://doi.org/10.3916/C57-2018-09
* To all the above, a formal question is added, for its correction by the authors, such as eliminating "Type of paper" from the beginning of the article.
* Personally, I have understood the entire exposition in English of the paper, but I am not a native English speaker. I consider it appropriate that someone from your journal check the writing in this language.
Best regards
Round 2
Reviewer 1 Report
Significant improvements have been achieved.
Reviewer 4 Report
Dear Authors,
Re: Manuscript “E-business strategy in developing countries: a framework and checklist for the small business sector”
Reviewer’s report:
The fundamental objective of the paper, which is to provide a framework for the development of electronic commerce in Nigerian companies, has been achieved. Obviously, if the companies interviewed for being at the beginning of this type of development cannot provide too much information, the framework obtained cannot be very powerful and concrete either. However, it can be valid in this context and to drive companies in it. However, for publication I recommend that the authors take into account the following issues:
- The discussion has improved somewhat. It would be convenient to go deeper into it and especially with the citations [22-24], so that it is clear what the current research contributes in relation to that of these citations.
- The authors state that the following paper has been very useful, and yet they do not cite it:
*Tong, A., Sainsbury, P., & Craig, J. (2007). Consolidated criteria for reporting qualitative research (COREQ): a 32-item checklist for interviews and focus groups. International Journal for Quality in Health Care, 19(6), 349-357. https://doi.org/10.1093/intqhc/mzm042
It is necessary to take it into account and expose aspects of its content so that the paper can gain in validity, and especially, transmit neutrality and reduced subjectivity in the work carried out for this study.
* To all the above, a formal question is added, for correction by the authors, such as the cut that occurs on page 13 (lines 415-417).
Best regards
